# Structure of a membrane-bound menaquinol:organohalide oxidoreductase

Lorenzo Cimmino[1,4], Américo G. Duarte [2,4], Dongchun Ni [3,4], Babatunde E. Ekundayo[3], Inês A. C. Pereira [2], Henning Stahlberg [3] ✉, Christof Holliger [1] ✉ & Julien Maillard [1] ✉

Organohalide-respiring bacteria are key organisms for the bioremediation of soils and aquifers contaminated with halogenated organic compounds. The major players in this process are respiratory reductive dehalogenases, corrinoid enzymes that use organohalides as substrates and contribute to energy conservation. Here, we present the structure of a menaquinol:organohalide oxidoreductase obtained by cryo-EM. The membrane-bound protein was isolated from *Desulfitobacterium hafniense* strain TCE1 as a PceA$_2$B$_2$ complex catalysing the dechlorination of tetrachloroethene. Two catalytic PceA subunits are anchored to the membrane by two small integral membrane PceB subunits. The structure reveals two menaquinone molecules bound at the interface of the two different subunits, which are the starting point of a chain of redox cofactors for electron transfer to the active site. In this work, the structure elucidates how energy is conserved during organohalide respiration in menaquinone-dependent organohalide-respiring bacteria.

Organohalide-respiring bacteria (OHRB) play an important role in the environment and in the natural halogen cycle, as they contribute to the degradation and detoxification of halogenated compounds that often accumulate in the environment due to natural and anthropogenic causes[1]. Organohalide respiration (OHR) represents an effective bioremediation approach[2,3]. In this anaerobic respiratory process, bacteria reductively dehalogenate organohalides and couple this reaction to ATP synthesis through chemiosmotic phosphorylation[4]. The key enzyme in this dissimilatory pathway is the reductive dehalogenase (RDase), a membrane-associated protein that is facing the P-side of the cytoplasmic membrane after transport via the Twin-arginine translocation (Tat) system. The active site of RDases has a cobalt corrinoid cofactor that mediates halogen elimination or substitution[5], and two additional iron-sulphur (FeS) clusters for intramolecular electron transfer[6]. RDases differ from other corrinoid enzymes, namely methyltransferases, and mutases, in that they likely catalyse the reduction of organohalogens through a nucleophilic attack of the

catalytically active Co(I) centre[7], while the latter ones normally involve the transfer of carbonyl groups[8].

The catalytic subunits (RdhA) display extensive sequence diversity across OHRB[9], most of them harbouring multiple copies in their genome as a consequence of horizontal gene transfer and gene duplication[10,11]. A sequence analysis of RdhA divides them into two major groups (Fig. 1a), which mostly reflect the dependency on quinones in the respiratory metabolism. There are bacteria that metabolically rely on menaquinones (MK), belonging mainly to the Firmicutes phylum that includes the genera *Dehalobacter* and *Desulfitobacterium*, and to the Proteobacteria such as *Sulfurospirillum*, and those that do not require quinones, from the Chloroflexi phylum, with *Dehalococcoides* and *Dehalogenimonas* as representative genera.

A highly conserved tetrachloroethene (PCE) reductive dehalogenase operon – *pceABCT* (Fig. 1b) is shared between *Desulfitobacterium hafniense* strain TCE1 and *Dehalobacter restrictus* strain PER-K23[12,13]. This operon codes for PceA, the catalytic subunit for reductive

[1]Laboratory for Environmental Biotechnology, Ecole Polytechnique Fédérale de Lausanne (EPFL), Lausanne, Switzerland. [2]Instituto de Tecnologia Química e Biológica António Xavier, Universidade NOVA de Lisboa, Oeiras, Portugal. [3]Laboratory of Biological Electron Microscopy, Ecole Polytechnique Fédérale de Lausanne (EPFL), and Department of Fundamental Microbiology, University of Lausanne, Lausanne, Switzerland. [4]These authors contributed equally: Lorenzo Cimmino, Américo G. Duarte, Dongchun Ni. ✉e-mail: henning.stahlberg@epfl.ch; christof.holliger@epfl.ch; julien.maillard@epfl.ch

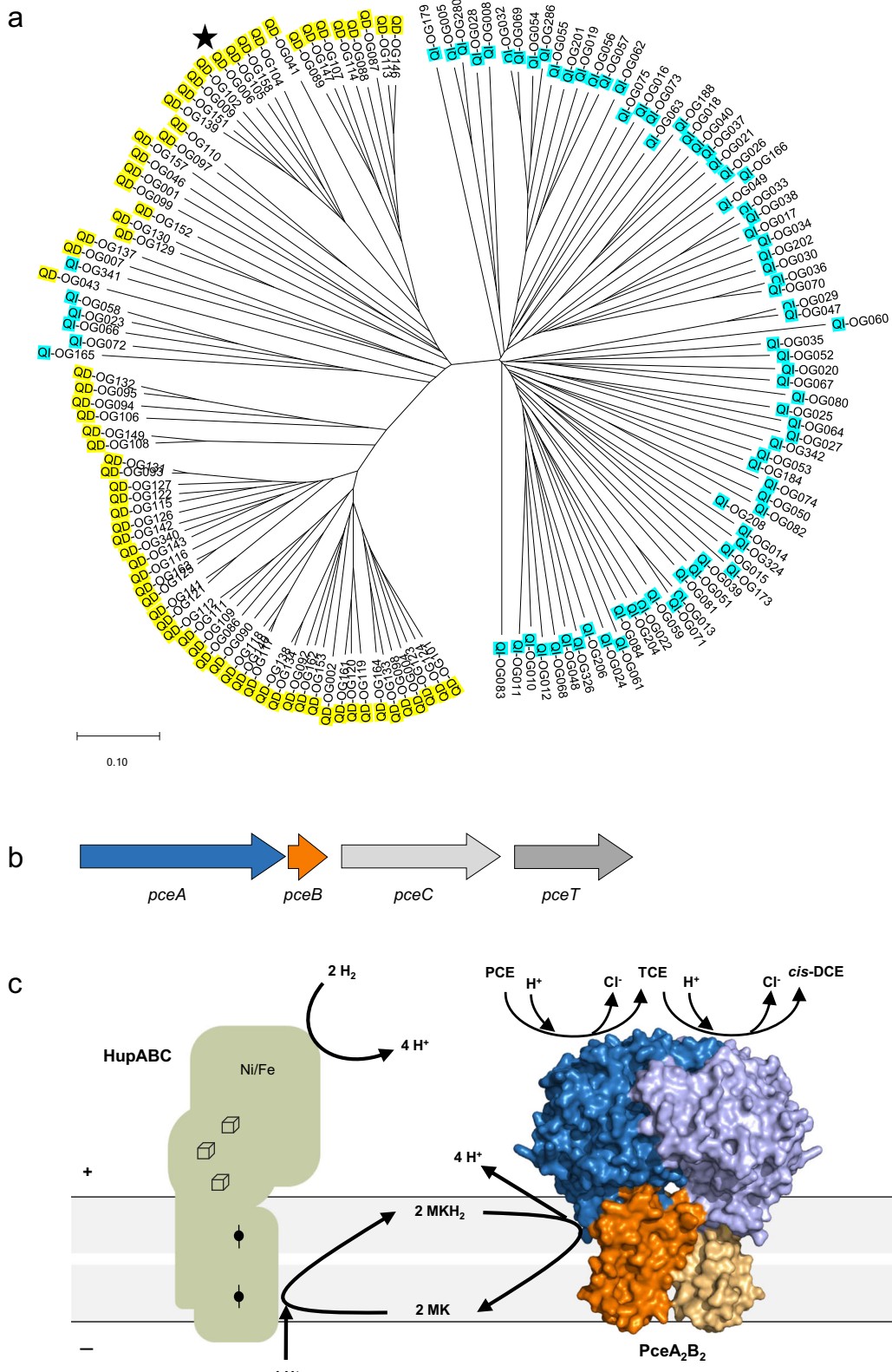

**Fig. 1 | Reductive dehalogenases. a** Phylogenetic tree of RdhA proteins. The star indicates PceA from *Desulfitobacterium hafniense* strain TCE1. **b** The *pce* gene cluster from *D. hafniense* strain TCE1[12]. **c** Model for the electron transfer chain in organohalide respiration by quinone-dependent bacteria growing with $H_2$ and PCE as electron donor and acceptor, respectively. The model displays the PceA$_2$B$_2$ reductive dehalogenase complex obtained in the present study. Legend: OG, orthologous group (as defined earlier[9]); QD: RdhA sequences from quinone-dependent bacteria; QI: RdhA sequences from quinone-independent bacteria; PCE, tetrachloroethene; TCE, trichloroethene; *cis*-DCE, *cis*-dichloroethene; Hup: uptake hydrogenase.

dechlorination, PceB an integral membrane subunit with three predicted transmembrane α-helices (TMH) and no evidence of cofactors[14], PceC a membrane-bound flavoprotein with still unknown function[15], and PceT, a molecular chaperone that binds to the Tat signal peptide of PceA and is involved in its folding before translocation[16]. In *D. hafniense*, this operon is located in a transposon that encodes a strong constitutive promoter, which drives PceA transcription[12,17], and recently it was shown that both *pceA* and *pceB* are transcribed at a significantly higher level when compared to *pceC* and *pceT*. Quantitative proteomic analysis of PceA and PceB proteins performed with *D. restrictus* and *D. hafniense* membrane extracts suggested a 2:1 ratio between the subunits PceA and PceB, while PceC was detected at a 50× lower abundance than PceA[18]. These results, and the absence of *pceC* in many OHRB, challenge the working model for electron transfer in *D. restrictus* that we have proposed in 2016, where PceC could be the missing redox protein between the quinol pool and PceA[19].

So far, two RdhA crystal structures have been solved: the catalytic subunit of the respiratory PCE reductive dehalogenase from *Sulfurospirillum multivorans* (SmPceA)[20] and the catabolic reductive dehalogenase from *Nitratireductor pacificus* (NpRdhA), a cytoplasmic NADPH-dependent enzyme involved in the degradation of 3,5-dibromo-4-hydroxybenzoic acid[21]. Despite some differences in the active site and the peripheral domains of both enzymes, the overall arrangement of the corrinoid- and FeS-containing domains appears to be similar between both structures[22]. In the SmPceA crystal structure, a homodimer of the catalytic subunit (PceA₂) is present, with a C-terminal domain that is highly flexible, which prevented determination of the structure in this region[20].

Investigation of the respiratory electron transfer chains from OHRB remains a challenge, due to their slow growth and low biomass yields, and to the fact that they are not genetically tractable. Based on genetics and proteomics results, OHRB from different phylogenetic groups likely harbour distinct sets of respiratory membrane complex architectures[3,6,14,23]. Bacteria among Chloroflexi are limited to using H₂ and acetate as energy and carbon source, respectively. These bacteria are MK-independent OHRB where a respiratory supercomplex was proposed to be responsible for energy conservation[24]. This supercomplex is composed of a hydrogenase (H₂-ase) module (HupLS), a membrane-bound reductive dehalogenase module (RdhAB), and three additional subunits (OmeAB-HupX) that belong to the complex iron-sulphur molybdoenzyme (CISM) superfamily[25]. The OmeB-HupX proteins were identified in several bioenergetic contexts, and their presence in the respiratory supercomplex of OHRB suggests that it may be responsible for energy conservation through a proton pumping mechanism, driven by the favourable reaction thermodynamics[26]. On the other hand, most of MK-dependent OHRB show a more versatile metabolism as they can couple the oxidation of H₂, formate or organic acids to the reduction of organohalides. In this case, the MK pool becomes reduced through a membrane-bound H₂-ase, a formate dehydrogenase or an alternative dehydrogenase, and menaquinol (MKH₂) oxidation is accomplished by a RDase complex through a proposed redox loop mechanism (Fig. 1c).

The molecular mechanism of how respiratory RDases operate is still unknown, and a matter of debate[3,14], namely regarding how electron transfer from the MKH₂ pool drives reductive dehalogenation. In vitro experiments show that MKH₂ is not able to transfer electrons directly to purified soluble RdhA enzymes, suggesting that an intermediate redox partner is missing[4]. A proteomic study in *S. multivorans* proposed a quinol dehydrogenase as a possible partner to transfer electrons from the menaquinol pool to PceA[27]. In *D. dehalogenans* and *D. hafniense*, the expression of a membrane-associated flavoprotein was highly up-regulated under OHR[28,29], but no evidence was obtained for its involvement in organohalide respiration.

Here, we show the structure of a menaquinol:organohalide oxidoreductase (MOOR) complex isolated from *D. hafniense* strain TCE1 as

a dimer of heterodimers (PceA₂B₂). Our study also unambiguously confirms the role of the RdhB subunit as a membrane anchor as proposed 25 years ago[30]. The MOOR complex presented here was obtained from the cell membrane fraction using a protocol developed to visualise PceA activity in native gels under strict anaerobic conditions. The structure of this complex reveals the presence of a quinone molecule at the interface between the PceA C-terminal domain and PceB, and provides the evidence of how energy is conserved during OHR, and how menaquinone-dependent respiratory reductive dehalogenases operate.

## Results

### *D. hafniense* expresses a PceA₂B₂ reductive dehalogenase complex

*D. hafniense* strain TCE1 encodes a single RDase (locus tag Desha-DRAFT_2412, for PceA), which is highly expressed when cells are cultivated with H₂ and PCE. Organohalide reduction activity has been reported in many OHRB using reduced viologen compounds as artificial electron donors[24,31–35], where this activity was detected either in the soluble or membrane fractions, or both[28,34,36]. Membrane proteins from strain TCE1 were solubilised with *n*-dodecyl-β-D-maltoside (DDM) and were analysed by Clear Native (CN)-PAGE, which showed a major band that migrates at an estimated molecular weight of 180 kDa (Fig. 2a). An in-gel activity assay was developed, where the PceA enzyme immobilised in the native gel uses reduced methyl viologen to reduce PCE, revealing a clear band on the otherwise dark red gel. This assay revealed an activity band for the membrane extract, with a similar molecular weight as the major band observed in CN-PAGE gel after Coomassie staining, as well as a smaller band migrating at ~140 kDa (Fig. 2b). A very faint activity band was also detected in the soluble fraction, also with ~140 kDa. The higher molecular weight band was extracted and analysed in a secondary dimension under denaturing conditions (Supplementary Fig. 1a, b), revealing two bands, one close to 60 kDa and another one close to 15 kDa, corresponding to PceA and PceB, whose presence was confirmed by LC/MS-MS analysis of the ~180 kDa band (Supplementary Fig. 2). A PceA₂B₂ complex would account for a predicted molecular weight (MW) of 142.7 kDa (including cofactors), which is in line with the observed size, considering the detergent micelles. The ~140 kDa can be attributed to a soluble PceA dimer that dissociates from the PceA₂B₂ complex, as it is present also in the soluble fraction (predicted MW = 119.7 kDa, with cofactors). These results indicate that a stable PceA₂B₂ complex is present in *D. hafniense* membranes, but which may dissociate during sample manipulation.

The membrane extract was fractionated by size-exclusion chromatography, enabling the purification of the PceA₂B₂ complex (Fig. 2c), and detergent replacement. This protein sample was processed for cryo-EM single-particle analysis (Fig. 2d). A subset of the data containing 34′078 particles was retrieved, from which the 3D refinement produced a C2 map with an overall resolution of 2.83 Å (Supplementary Fig. 3a, b). High-quality cryo-EM density maps were used to build atomic models de novo for both PceA and PceB subunits (Supplementary Fig. 3c), which form a large supramolecular complex.

### Cryo-EM structure of the *D. hafniense* PceA₂B₂ complex

The large PceA subunit has a globular shape structured with α- and β-folds, while PceB has three TMH predicted to be fully integrated into the lipid bilayer, thus acting as the membrane anchor for the complex. These subunits are assembled as a dimer of heterodimers (PceA₂B₂), with a C2 symmetry (Fig. 3a–c). Similar to SmPceA and NpRdhA, *D. hafniense* PceA (DhPceA) has a cobalamin active site and binds two [4Fe-4S] clusters[20,21]. Unexpectedly, electron density for two menaquinone molecules was also observed, each at the interface between PceA and PceB subunits (Fig. 3d–f). Finally, density for a dichloroethene (DCE) molecule, the final product of PCE dechlorination[37], was observed 5.3 Å away from each of the two active site cobalt atoms (Fig. 3g).

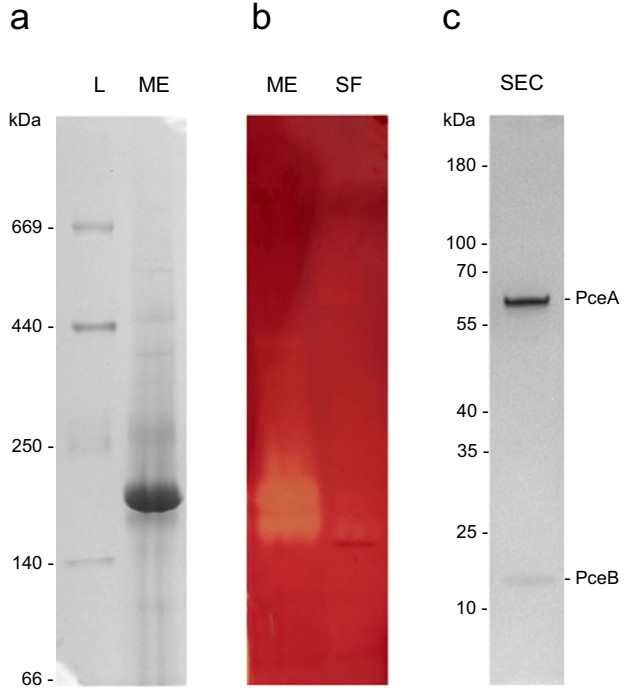

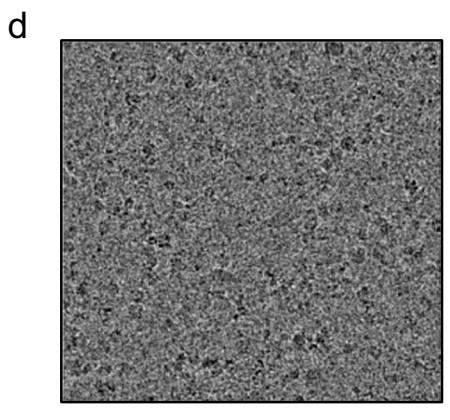

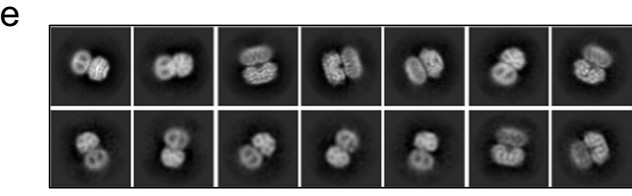

**Fig. 2 | Reductive dehalogenase complex sample characterisation. a, b** One representative Clear-Native PAGE gel of the membrane extract (ME) and soluble fraction (SF): **a** stained by Coomassie and **b** stained by the in-gel enzymatic activity. Lanes represented in **a** and **b** were issued from the same gel and cut for staining. This experiment has been performed three additional times with similar results. **c** SDS-PAGE of the major protein peak after size-exclusion chromatography (SEC) of the ME sample. This experiment has been performed once. **d** One representative cryo-EM micrograph (out of 13783 images), scale bar: 100 nm. **e** Selected class averages of particles of the complex, scale bar: 20 nm. Source data are provided as a Source Data file.

In terms of topology organisation, PceA has 19 α-helices and 13 β-sheets (Supplementary Fig. 4a), which fold in a compact arrangement, with the cobalamin cofactor buried inside, while the FeS clusters are slightly exposed in the protein C-terminal domain. In the interface between the two subunits, the quinone molecule is fenced between two side by side α-helices of PceA (α17 and α18), at the edge of the cytoplasmic membrane facing the P-side, with two TMH from PceB further enclosing the quinone (Fig. 3f).

DhPceA has only 28 % amino acid sequence identity with SmPceA (Supplementary Fig. 5a) and the superimposition of their structure retrieves a RMSD value of 3.095 Å, reflecting a somewhat different protein peripheral topology (Supplementary Fig. 5a, b) and the lack of structure in the SmPceA C-terminus[20], due to the absence of the cognate PceB subunit. Structural alignment between DhPceA and NpRdhA is only partial since NpRdhA is a cytoplasmic NADPH-dependent enzyme and more distantly related to respiratory reductive dehalogenases (Supplementary Fig. 5a, c). Nevertheless, in all three structures, the organisation of the active site and FeS clusters is similar, since they all superimpose with minimal shifts (Supplementary Fig. 5b, c). As observed for SmPceA and NpRdhA, the two active sites in $PceA_2B_2$ are far away from each other, with no cofactor between them, which suggests that the complex has two independent catalytic sites, one per heterodimer, as it was already proposed for *S. multivorans* PceA dimer. However, possible cooperativity between both binding sites cannot be excluded.

In our structure, the catalytic site is 8.3 Å apart from the proximal FeS cluster, that is 9.9 Å away from the distal FeS cluster, which in turn is aligned with the menaquinol hydroxyl groups with distances shorter than 12 Å (Fig. 3g). This organisation forms a conductive redox path between the menaquinol pool and the active site. PceA interacts with PceB at a ~ 60° angle, and due to the complex symmetry, the first TMH of the membrane subunits are close to each other, crossing close to the P-side of the membrane (Fig. 3d). Overall, the $PceA_2B_2$ subunits display relatively intricate contact surfaces, where the largest contact area is between the two catalytic subunits (3016 Å², Fig. 3a). PceA helix α5 from each subunit establishes interactions with the α6 helix from the other subunit and with helix α11 from both PceA subunits, while helices α7 from both subunits prolong each other (as shown by the dashed line in Supplementary Fig. 4c). At the top of the structure, the β1 and β2 strands and their connecting loop, also contribute to the tight assembly of the PceA monomers (Fig. 3d and Supplementary Fig. 4c). Based on the calculated charge density, the PceA-PceA interaction is mostly electrostatic since the dimer interface has opposite charges in each half of the subunits (Supplementary Fig. 6a). As mentioned, the PceB subunits interact with each other through the contact of TMH1, with a contact area of 489 Å², mostly supported by hydrophobic interactions (Supplementary Fig. 6b). In each heterodimer, the C-terminus of PceA binds to the PceB TMH2-TMH3 connecting loop via electrostatic interactions (Supplementary Fig. 6b, c). Additionally, there is a direct interaction between PceA of one heterodimer and the N-terminal helix of PceB from the other heterodimer, with an average surface area of 166 Å² (Fig. 3d). All these structural features support the idea that the $PceA_2B_2$ complex is a physiological dimer, but with independent electron pathways from the quinone pool to the active sites.

## PceA active site and channels for substrate/product diffusion

At the PceA active site of *D. restrictus* (and most likely also at that of *D. hafniense* strain TCE1), there is a cobalt-containing corrinoid held in a base-off conformation, as identified earlier by EPR spectroscopy[38] and observed previously in the SmPceA and NpRdhA structures[39]. Several polar residues, namely N68, H395, K399, N454, H461, and S479 are at hydrogen-bond distance (< 7 Å) from polar edges of the catalytic ring, supporting its binding to the polypeptide chain (Supplementary Fig. 7). The substrate binding cavity is surrounded by non-polar and

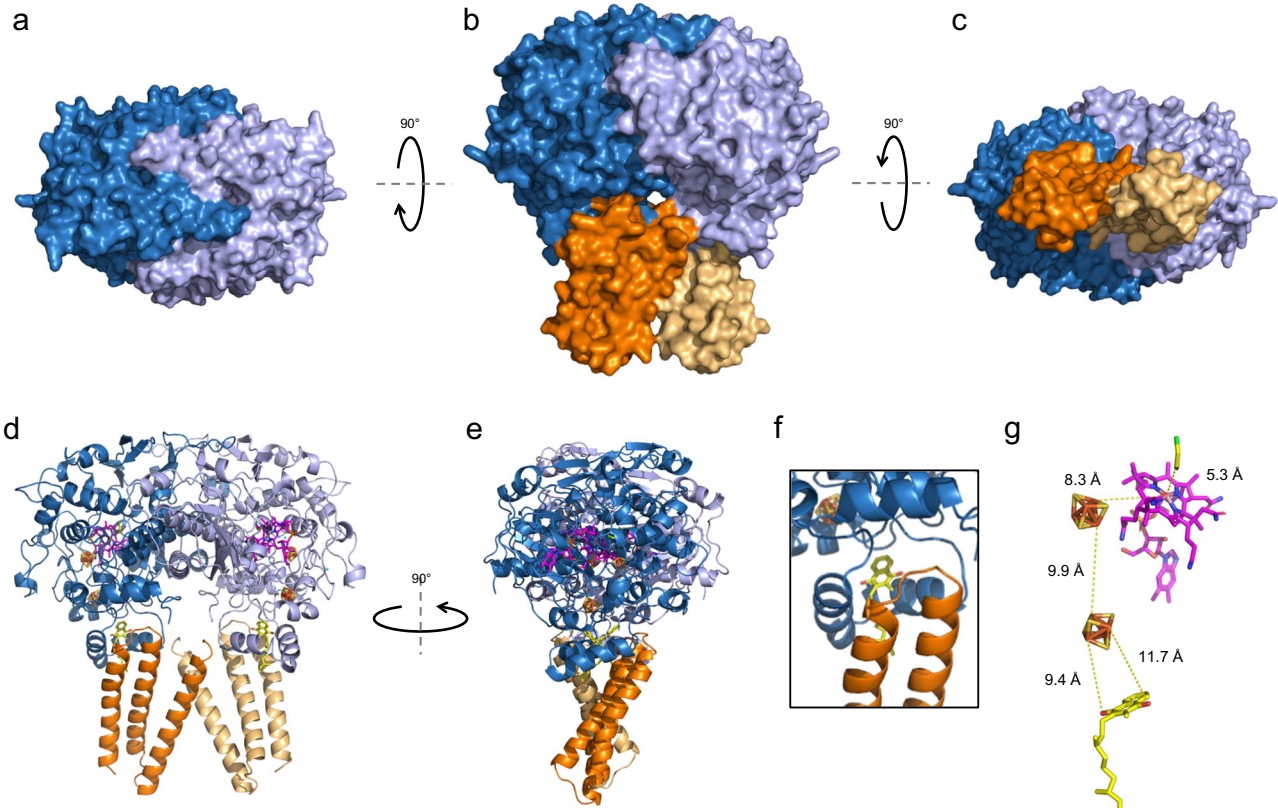

**Fig. 3 | Cryo-EM structure of the PceA$_2$B$_2$ complex from *D. hafniense* strain TCE1.** Representation of the surface of the complex from **a** the top, **b** the front and **c** the bottom view. Cartoon representations of **d** the front and **e** the side view of the protein complex. **f** Detail of the menaquinone-binding pocket in front view. **g** Cofactor arrangement in one PceAB heterodimer indicating the estimated distances between the cofactors. Colour code: PceA subunits are depicted in blue and light purple, PceB subunits in orange and beige; the menaquinone is in yellow, both [4Fe-4S] clusters in orange/yellow, the cobalamin in magenta and the dechlorination product (*cis*-1,2-dichloroethene) in yellow/green.

aromatic amino acids (Fig. 4a), some of which were shown to be relevant for the catalytic mechanism in SmPceA[20]. There, a tyrosine residue (Y246) was proposed to be involved in this mechanism by mediating proton transfer, and an arginine (R305) was proposed to be involved in the stabilisation of a tyrosyl intermediate, since site-direct mutagenesis of these residues led to a decrease or abolishment of the enzymatic activity[5,20,39]. In DhPceA, these amino acids are structurally conserved as Y337 and R396, respectively (Fig. 4a). A comparison of DhPceA and SmPceA active sites reveals that some residues of the substrate cavity are replaced by others and some are in a different orientation. The striking differences are SmPceA W56, which is structurally replaced by F127 in DhPceA, adopting a slightly shifted position, while SmPceA T242 and Y382 are structurally replaced by V333 and P477 in the structure of DhPceA. Additionally, SmPceA Y102 located in a loop following the structurally conserved α5, is replaced by F166 in DhPceA in a different orientation (Fig. 4a and Supplementary Fig. 5d)[20].

Using CAVER Analyst 2.0[40] two channels are predicted for the PceA catalytic subunit of *D. hafniense* (Fig. 4b, c), in contrast to the described "letter box" channel observed in PceA of *S. multivorans*[20]. One of these channels goes from the active site towards the top of the structure and includes the DCE product molecule (as depicted in Fig. 4c), while the other channel extends from the side of the corrinoid ring, opposite to the proximal FeS cluster, towards the surface (depicted in magenta). These channels are less than 40 Å in length and their narrowest section has a 0.95 Å radius. While this section appears too narrow for the diffusion of chlorinated ethenes, it must be noted that this analysis is static and does not consider the flexibility of the channel upon substrate transfer to and out of the active site. Both channels connect the active site to the outside solution

(Supplementary Fig. 8), and display a hydrophobic inner surface adequate for substrate and product diffusion.

## Key residues in the MK pocket support proton transfer to the P-side

In the PceA C-terminal domain (S486-Q551), two α-helices predicted to be embedded in the membrane (α17 and α18, Supplementary Fig. 4a, b) lay close to each other at a 53° angle, and bind a menaquinone molecule between them (Fig. 3d–f; Fig. 5a). These two helices have hydrophobic and polar residues, namely S486, W487, K489, H495, A498, R499, A511, D515 and Y520, which can stabilise the menaquinone and support its redox chemistry (Fig. 5b). Together with PceB TMH2 and TMH3, they form a four-helix bundle that encases the menaquinone molecule (Fig. 5a). A similar quinone-binding site has been also observed in the quinone-interacting NrfH subunit from the NrfA$_4$H$_2$ nitrate reductase complex[41]. In PceB, a group of hydrophobic residues (mainly Phe residues) stretches deeply into the lipid layer, and likely stabilises the quinone isoprene tail (Fig. 5c), which is only partially resolved in the structure.

Interestingly, in the quinone-binding site, three polar amino acids seem to stabilise the quinone head, due to their proximity to the molecule polar groups: K489 (at 3.7 Å), H495 (2.5 Å), R499 (5.4 Å) and D515 (4.5 Å) (Fig. 5b). The PceB loop between TMH2 and TMH3, on the P-side of the membrane, displays a sequence of four consecutive protonatable residues, among which E61 and E63 have been proposed to play a role in the RdhA-RdhB interaction[14]. In contrast, we think that these residues, together with conserved residues in PceA (H495, D515, Y520), could be involved in the transfer of protons from the MKH$_2$ oxidation to the P-side of the membrane (Fig. 5d). Primary sequence

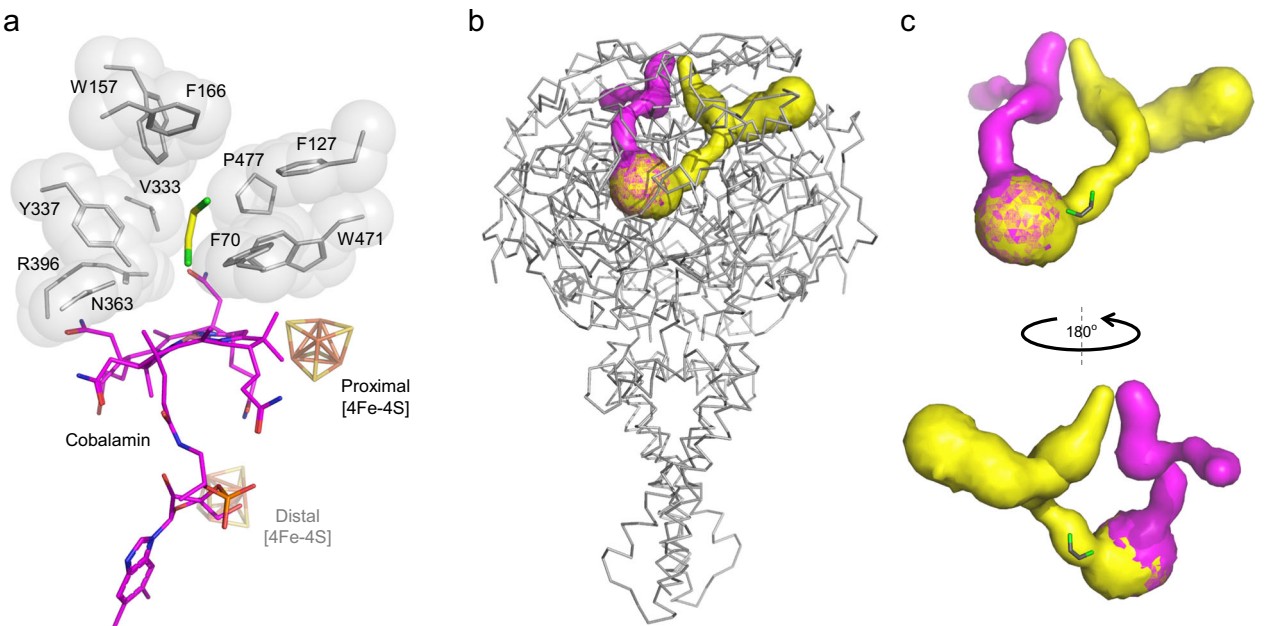

**Fig. 4 | Active site and channels in the PceA₂B₂ complex. a** The active site of PceA, and **b, c** the possible channels (in yellow and pink) for substrate and product diffusion in and out of the active site.

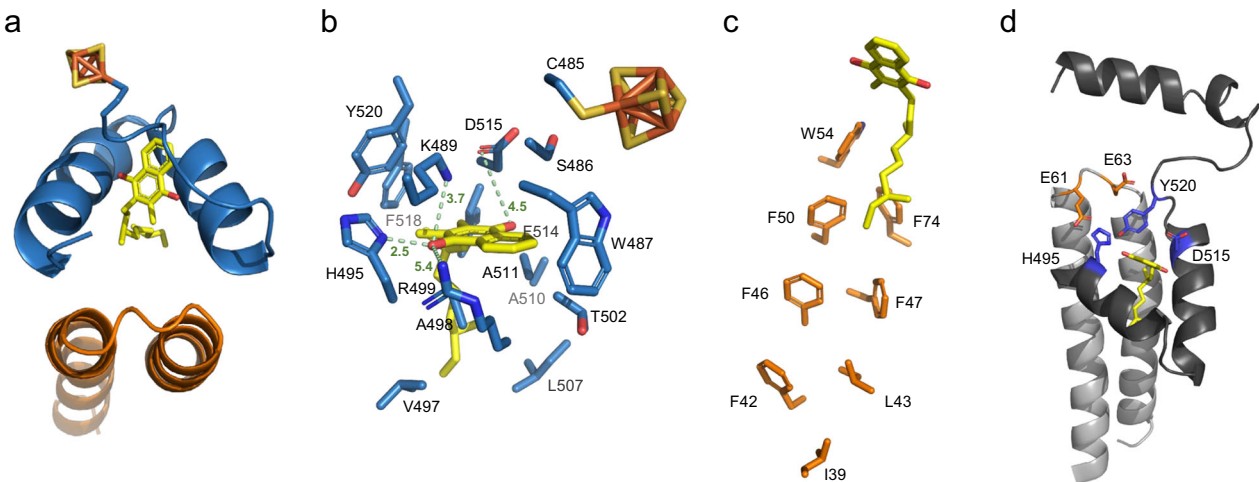

**Fig. 5 | The menaquinone-binding pocket of the PceAB heterodimer. a** Cartoon representation of the four-helix bundle involved in binding the menaquinone. **b** PceA and **c** PceB amino acids likely involved in binding the quinone head and tail (here resolved with only two isoprene subunits), respectively. **d** Cartoon representation of the menaquinone-binding pocket highlighting conserved protonatable residues in PceA (in blue) and in PceB (in orange) that could be involved in proton transfer.

alignments of RdhA and RdhB sequences show that some of these amino acids are highly conserved among RDases from bacteria that use quinones in their metabolism, but also in organisms that do not use quinones (Supplementary Fig. 9). Analysis of the PceB structure shows that there are no additional conserved protonatable residues in the TMHs, supporting the importance of the two glutamate residues for energy conservation.

## Discussion

In MK-dependent OHRB, quinones have been proposed to play a central role in the respiratory chain by supporting the reduction of organohalide compounds and being involved in proton transfer to generate the protonmotive force (*pmf*) that enables ATP synthesis[4,6]. By monitoring PCE reductive dehalogenase activity in native gel electrophoresis combined with protein identification through mass

spectrometry, a complex was identified, extracted and purified from the membranes of *D. hafniense* strain TCE1, and its structure solved by cryo-EM. The structure shows a tetrameric complex (PceA₂B₂) with two independent corrinoid-containing catalytic sites, each of them connected to a menaquinone-binding site by two [4Fe-4S] cofactors, in agreement with previous experiments with whole-cells extracts demonstrating the role of the quinone pool in electron transfer to the catalytic subunit[36,42]. Remarkably, a menaquinone molecule is observed at the interface of the catalytic PceA subunit and the membrane anchor PceB, revealing that both subunits are involved in binding the quinone and explaining why an earlier study on *D. restrictus* PceA, without PceB, failed to detect electron transfer from reduced quinones to PceA[38]. The lack of the PceB subunit in the *S. multivorans* PceA structure also explains why its C-terminal region appears unstructured[20]. Our results suggest that the integrity of the

PceA$_2$B$_2$ complex is sensitive to the membrane extraction and protein purification process. Moreover, as observed for SmPceA, DhPceA displays in its active site the two conserved residues essential for organohalide reduction (Y337 and R396). In addition, DhPceA has two hydrophobic channels for substrate and product diffusion, given the chemical nature of PCE and its dechlorination product cis-DCE.

Reductive dehalogenases can use several substrates such as chlorinated and brominated compounds, and perform single or multiple halide removal upon catalysis[14]. SmPceA for instance can dehalogenate chlorinated ethenes and propenes, brominated ethenes and phenols[39,43], while DhPceA and highly similar enzymes[13] are more selective and use only chlorinated ethenes and ethanes as substrates[31,37,44]. The substrate specificity can derive from the size and chemical nature of the catalytic cavity, and most certainly also from the channels that connect the active site to the enzyme exterior. Regarding substrate specificity, the W432F variant of *Desulfitobacterium dichloroeliminans* 1,2-dichloroethane reductive dehalogenase (corresponding to DhPceA W471) decreased the dechlorination rate of PCE but improved that of bulkier compounds such as tribromoethene, indicating that the wider substrate cavity of the variant hampers the correct positioning of smaller substrates[5]. The same study showed the opposite effect with the variant T294V (corresponding to T242 in SmPceA and V333 in DhPceA), with the same explanation. Therefore, a valine at this position in DhPceA may be one of the factors that restrict its substrate specificity in comparison to that of SmPceA. Interestingly, a single mutation (I481T) in TceA from *D. mccartyi* seemed to confer an increased rate of vinyl chloride dechlorination, and structural homology modelling predicted that another mutation is located near the active site and may alter the substrate access channel, thus possibly enhancing dechlorination[45,46]. It has also been shown that a few amino acid substitutions can affect the reaction mechanism and substrate affinity in chloroform-reducing reductive dehalogenases[47]. Modifications of the channels can have a drastic impact on the enzymatic activity and can be seen as a tool for enhancement of relevant enzymes[48]. In DhPceA, the predicted hydrophobic channels have constricted sections, which may also explain the selectivity for two-carbon chlorinated substrates. Additionally, the presence of a cis-DCE molecule in one of the channels near the active site is in line with the physiological observation that PCE is converted to cis-DCE without any accumulation of TCE[44]. It is likely that TCE is readily dechlorinated and does not diffuse out of the active site, but no evidence for this was obtained.

From a thermodynamic point of view, it has been suggested that MKH$_2$ cannot transfer electrons to RDases based on its redox potential ($E^{0'} = -74$ mV) that is substantially higher than the redox potentials determined for the *D. restrictus* PceA active site ($E^{0'}_{Co(II)/Co(I)} = -350$ mV) or for the two [4Fe-4S] clusters (~$-480$ mV)[38]. However, electron transfer to a certain redox site should not be excluded simply because its midpoint redox potential is less favourable. In the case of reductive dehalogenation, substrates have much higher redox potentials (from +240 to +580 mV[14]), which make their reduction by MKH$_2$ favourable and drive electron transfer[49]. MK-dependent OHRB couple H$_2$ oxidation ($E^{0}_{H+/H2} = -432$ mV) to organohalide reduction through membrane-bound H$_2$-ases that have a di-heme cytochrome $b$ membrane subunit typically with a MK-binding site on the N-side of the membrane. The hydrogenase complex receives electrons from the P-side of the membrane from H$_2$ oxidation, while protons are consumed from the cytoplasm (Fig. 1). We show here that MOOR complexes have their quinone-binding site facing the P-side of the membrane and have conserved protonatable amino acids to liberate protons to this same side. This classifies the MOOR complexes as electroneutral[50], since the electrons and protons from MKH$_2$ oxidation are released on the same side of the membrane. However, the MOOR and hydrogenase complexes form a quinone/quinol redox loop that results in the establishment of a *pmf* (Fig. 1). This redox loop results in

the transfer of 4H$^+$ from the cytoplasm to the periplasm per PCE molecule converted, due to the position of the two quinone/quinol binding sites on opposite sides of the membrane. The 4 scalar H$^+$ released from oxidation of two H$_2$ molecules are used up in the PCE reaction.

MK-independent OHRB lack the genes for quinone biosynthesis[51], and their respiratory chain seems to be restricted to a modular supercomplex, as described in *D. mccartyi*, composed by a H$_2$-ase, a respiratory RDase module and a OmeA/OmeB/HupX module, which is possibly involved in energy conservation[24,25]. The absence of quinones in these organisms suggests that energy conservation is based on a proton pumping mechanism, which is likely dependent on the existence of proton channel(s) within the supercomplex. Since the protonatable residues identified in the MK-binding site of DhPceA$_2$B$_2$ are conserved in RDases from MK-independent OHRB, it is possible that these residues may also play a role in the transport and/or release of protons to the P-side of the membrane in the supercomplex, even in the absence of MKH$_2$ oxidation.

Among MK-dependent OHRB, the DhPceA$_2$B$_2$ complex is representative of the minimal respiratory RDase module that is most likely capable of delivering electrons from MKH$_2$ to the catalytic subunit. This is likely also valid for *rdh* gene clusters that include a gene coding for the membrane-bound flavoprotein RdhC, in addition to the *rdhA* and *rdhB* minimal gene set. However, the discovery on a genomic level of hybrid RDases which lack the N-terminal Tat signal peptide but harbour transmembrane helices that resemble that of RdhB (including the conserved EXE motif[52]) may indicate that these fused RDases are most likely structurally related to the PceA$_2$B$_2$ complex, although their physiological function and enzymatic activity remain to be assessed.

In conclusion, the present work describes the structure of a respiratory reductive dehalogenase complex, where the PceA catalytic subunit is anchored to the membrane via PceB, and together form a quinone binding site. Menaquinol is likely oxidised on the P-side of the membrane, and electrons are thought to be transferred to the catalytic cobalamin site through two [4Fe-4S] centres. PceA has a set of hydrophobic channels that are likely determinants for substrate selectivity. This structure provides evidence for the molecular mechanism used by MK-dependent OHRB to conserve energy, revealing that the MOOR complex is electroneutral but contributes to the *pmf* via the positive balance in the stoichiometry of protons released at the MKH$_2$ oxidation site and those consumed by the substrate turnover.

## Methods
### Bacterial growth conditions
*Desulfitobacterium hafniense* strain TCE1 (DSM 12704) was cultivated anaerobically at 30 °C under constant agitation (100 rpm). The medium was prepared as previously described[53], and autoclaved after replacing the head space with a mixture of N$_2$/CO$_2$ (80%/20%). The medium was supplemented with cyanocobalamin at the final concentration of 50 µM and acetic acid at 5 mM as the carbon source. The flasks head space was replaced by H$_2$/CO$_2$ (80%/20%) to provide H$_2$ as an electron donor, and 1% (v/v) of a 2 M PCE solution prepared in hexadecane was added as electron donor to the medium in a biphasic system[54], leading to an estimated aqueous PCE concentration of 0.4 mM. The culture flasks were inoculated with 5% (v/v) of a preculture prepared in the same conditions.

### Cell harvest and fractionation
All subsequent steps were performed in an anaerobic glove-box (Coy Laboratory). Biomass from approximately 2.5 L of culture media was harvested after 3–4 days of cultivation, corresponding to the late growth phase. After removal of the PCE-containing organic phase from the culture media, cells were harvested using Harvest Liners (Beckman Coulter) by centrifugation at 12000 × *g* and 4 °C for 20 min. A yield of

approximately 0.4 g (wet weight) of biomass per L culture was typically obtained. Cells were washed with 50 mM Tris–HCl (pH 7.5), transferred to cryotubes and collected by centrifugation at 10000 × g for 3 min, the supernatant of which was discarded. The biomass pellet containing cryotubes were stored in anaerobic serum flask at −80 °C until further use.

Frozen cells were thawed and suspended in 50 mM Tris-HCl (pH 7.5) supplemented with a few crystals of DNase I (Roche) and protease inhibitors (PI) (cOmplete mini, EDTA-free, Roche). To ensure anaerobic conditions, the French press chamber was first rinsed with oxygen-free 50 mM Tris-HCl (pH 7.5) with 100 mM dithiothreitol (DTT). Cell disruption was made with three rounds of French press at 1000 psi (6.9 MPa) and 4 °C. To avoid air contamination of the sample, the inlet and outlet of the French press were connected to anaerobic tubes. After cell disruption the suspension was immediately transferred to the anaerobic glove-box. Cell debris were removed by centrifugation at 500 × g and 4 °C for 20 min in gas-tight tubes and the resulting supernatant was centrifuged at 90000 × g for 90 min at 4 °C, yielding the soluble fraction and the membrane pellet. The pellet was resuspended in oxygen-free 50 mM Tris-HCl (pH 7.5) supplemented with PI, and submitted to ultracentrifugation as before. The membrane pellet was resuspended in the same buffer and the protein concentration was determined spectroscopically with the BCA protein assay kit (Thermo Fisher Scientific). The membrane fraction was supplemented with n-dodecyl-β-D-maltoside (DDM) at a DDM:protein (w/w) ratio of 10:1 and with 10% glycerol (final concentration). The sample was stirred overnight at 4 °C under anaerobic conditions. DDM-solubilised membrane proteins were recovered by ultracentrifugation as before. If required, the membrane protein samples were concentrated using 2-mL ultrafiltration devices (50 kDa cut-off, Amicon). Protein concentration was determined with the BCA protein assay kit.

### Clear-Native and SDS-PAGE
Clear-Native (CN) gels were hand-cast with a 5 to 15% (w/v) acrylamide/Bis-acrylamide gradient in Bis-Tris buffer[55]. Aliquots of 20-50 μg of total protein were loaded in the gel and separated by applying 6 mA constant current per gel and limiting the voltage to 300 V for 3 h 30 min. Replicate samples were loaded for total protein analysis (Coomassie staining) and for the in-gel enzymatic assay (see below). In the latter case, gel-electrophoresis was performed under strict anaerobic conditions, inside the glove-box. SDS-PAGE was performed with 12% acrylamide gels loading 50 μg of total protein or using a piece from CN-PAGE gels as sample. For gels, source data was provided as a Source Data file.

### LC-MS/MS analysis
Shotgun mass spectrometry (MS) analysis was performed on an Orbitrap Exploris 480 mass spectrometer (Thermo Fisher Scientific) coupled to a nano-UPLC Dionex pump. For liquid chromatography (LC)-MS/MS analysis, Trypsin digested samples were resuspended in 30-60 μL of a mobile phase (solvent A: 2% acetonitrile (ACN) in water, 0.1% formic acic (FA)) and then separated by reversed-phase chromatography using a Dionex Ultimate 3000 RSLC nanoUPLC system on a home-made 75 μm ID × 50 cm C18 capillary column (Reprosil-Pur AQ 120 A˚, 1.9 μm) in-line connected with the MS instrument. Peptides were separated by applying a non-linear 150 min gradient ranging from 99% solvent A to 90% solvent B (90% ACN and 0.1% FA) at a flow rate of 250 nL/min. For spectral library and charge state determination of the peptides from PceA and PceB of *D. hafniense* strain TCE1, the MS instrument was operated in data-dependent mode (DDA). Full-scan MS spectra (300-1500 m/z) were acquired at a resolution of 120'000 at 200 m/z. Data-dependent MS/MS spectra were recorded followed by HCD (higher-energy collision dissociation) fragmentation on the ten most intense signals per cycle (2 s), using an isolation window of 1.4 m/z. HCD spectra were acquired at a resolution of 60'000 using a

normalised collision energy of 32 and a maximum injection time of 100 ms. The automatic gain control (AGC) was set to 100'000 ions. Charge state screening was enabled such that unassigned and charge states higher than six were rejected. Precursors intensity threshold was set at 5'000. Precursor masses previously selected for MS/MS measurement were excluded from further selection for a duration of 20 s, and the mass exclusion window was set at 10 ppm.

For peptide identification, the following settings were used: enzyme: Trypsin; missed cleavages: 2; precursor mass tolerance: 10 ppm; fragment mass tolerance: 0.2 Da; minimum charge: 2; maximum charge: 5; fixed modifications: Carbamidomethyl (Cys); variable modifications: Oxidation (Met), Phosphorylation (Ser, Thr, Tyr). False discovery rate (FDR) was calculated based on the target/decoy database and peptides as well as proteins with FDR threshold of 1% were chosen as true positive hits. Data analyses were performed using Skyline (version 21.1.0.278, MacCoss lab, University of Washington, USA), an open source software tool application for data processing and proteomic analysis.

### Additional analytical methods
Protein detection was performed either by Coomassie or Silver staining with standard procedures.

### In-gel PCE reductive dehalogenase enzymatic activity assay
PceA reductive dehalogenase activity was assayed in anaerobic conditions at room temperature (inside a Coy Laboratory glove-box with a 97%/3% (v/v) $N_2/H_2$ atmosphere). After protein separation by CN-PAGE, the gel piece (corresponding to up to three lanes) was vertically placed in a 500-mL anaerobic serum flask with 10 mL oxygen-free 50 mM Tris-HCl buffer (pH 7.5). The flask was removed from the glove-box, gas-exchanged for $N_2$ and placed back into the glove-box. The buffer was removed and replaced by 0.5 mL of a 250 mM methyl viologen (MV) solution, that was previously reduced with 1 g of zinc powder (Sigma-Aldrich). The gel was incubated for approximately for 15–30 min, until a homogeneous blue staining was obtained. After removing the excess MV solution, 1 mL of an anaerobic 10 mM PCE solution in ethanol was added to the gel and the assay was incubated at room temperature until colourless bands appeared on the gel due to the MV oxidation by the reductive dehalogenase activity. When bands were clearly visible, 5 mL of a 300 mM 2,3,5-triphenyltetrazolium chloride solution was added producing homogeneous and oxygen-resistant red staining of the MV-reduced portions of the gel, leaving the bands with enzymatic activity colourless.

### Negative staining EM
Before analysis by negative staining EM, the suitable sample buffer was tested by diluting 20× the membrane protein extract solubilized in DDM (total concentration of 2.84 mg/mL), in four different buffers: Buffer 1: 25 mM HEPES-NaOH pH: 7.5, 150 mM NaCl, 0.1% DDM; Buffer 2: 25 mM HEPES-NaOH pH: 7.5, 0.1% DDM, 5% glycerol; Buffer 3: 25 mM HEPES-NaOH pH: 7.5, 150 mM NaCl, 0.005% Lauryl Maltose Neopentyl Glycol (LMNG); and Buffer 4: 25 mM HEPES-NaOH pH: 7.5, 0.005% LMNG, 5% glycerol.

For negative staining EM, 3 μL of diluted sample was applied on glow discharged 400 mesh copper grids on carbon film. The grids were incubated with 2% (w/v) uranyl acetate for 30 s for staining. Grids were imaged on the Philips CM 100 transmission electron microscope (TEM). The results from screening showed that the sample in Buffer 3 produced a homogeneous population of mono-dispersed particles suggesting that this buffer condition would be optimal for further EM analysis.

### Protein purification
To purify the RDH complex for cryo-EM, size exclusion chromatography was performed. The membrane protein extract was pre-

equilibrated in Buffer 3 in order to reduce detergent concentration and loaded in a Superose 6 Increase 10/300 GL column (Cytiva) at a flow rate of 0.5 mL/min on an ÄKTA Go FPLC system (Cytiva). Fractions containing the RDH complex were pooled, concentrated to 1 mg/mL using a 30 kDa cut-off centrifugal concentrator (Millipore), and their purity analysed by SDS-PAGE.

## Cryo-EM sample preparation and data acquisition
Cryo-EM grids were prepared by applying 3.5 μL of concentrated sample at 1.2 mg/mL to glow discharged 400 mesh R1.2/1.3 UltrAuFoil grids (Quantifoil Micro Tools GmbH). The sample was adsorbed for 30 s on the grids followed by plunge freezing in liquid ethane using a Vitrobot Mark IV plunge freezer (Thermo Fischer Scientific). The cryo-grid was pre-screened on a Glacios TEM (ThermoFischer Scientific) and then transferred to a Titan Krios G4 transmission electron microscope (ThermoFischer Scientific). The microscope was equipped with a cold-FEG and a Falcon IV camera at an operating voltage of 300 kV. 13'783 cryo-EM movies were recorded using data automation software EPU (Thermo Fischer Scientific) with a physical pixel size of 0.51 Å. Defocus ranged from -1.0 μm to -2.2 μm and a calibrated dose of 60 e/Å$^2$ in total was applied.

## Cryo-EM image processing
Data Analysis was performed using the cryo-EM image processing software cryoSPARC v3.3[56]. Movies were motion-corrected, dose-weighted, and exported as mrc files. The data collection statistics and processing flowchart are shown in Supplementary Fig. 3 and Supplementary Table 1.

Cryo-EM micrographs were imported into cryoSPARC v3.3 and a patch-based contrast transfer function (CTF) estimation was applied. Blob picking was applied, followed by 2D classification to create a 2D template for subsequent template picking. By performing template picking, 1'049'505 particles were automatically picked. After 4 rounds of 2D classification as well as 2D rebalancing, 83'941 particles were selected for subsequent 3D reconstruction.

Ab-Initio & Hetero Refinement yielded an optimal 3D class with 55'431 particles. In addition, a subset of particles was classified into three classes by 3D classification. The final best class containing 34'078 particles were then implemented for 3D refinement with C2 symmetry imposed, resulting in a 3D map with a global resolution of 2.83 Å. Resolution was estimated by standard method FSC with a 0.143 cut-off value. Local resolution was calculated by Local Resolution Estimation in cryoSPARC.

## Model building and refinement
The obtained cryo-EM maps were of high resolution and quality of density and could be used for de novo modelling. A C-α tracing model was built manually in coot v0.9.4.1[57]. The sequences of both PceA and PceB were identified with high confidence and matched to the density. After multiple rounds of manual building, an atomic model of the PceA$_2$B$_2$ complex was established. The structural geometry and rotamers, etc. were optimised using real.space.refine in Phenix v1.19.2-4158[58]. The relevant statistics are in Supplementary Table 1. Figures were rendered with PyMOL (The PyMOL Molecular Graphics System, Version 2.0 Schrödinger, LLC) and UCSF ChimeraX v1.4[59].

## RdhA and RdhB sequence analysis
The RdhA sequences were selected from the published database RDaseDB[9] by considering the following two criteria: (i) the presence of a cognate RdhB coding sequence in the direct vicinity of the *rdhA* gene, and (ii) a sequence divergence > 5% than any other sequence in the selection. A list of 155 RdhA sequences was obtained and their cognate RdhB sequences extracted from GenBank. The original annotation of orthologous groups (OG) was used as defined in RDaseDB. Both RdhA and RdhB sequences were aligned with

ClustalX (v.2.0.12)[60]. The phylogenetic tree was built with MEGA X[61]. In order to identify conserved amino acids in both subunits, T-Coffee Expresso web server[62] was used to align the sequences of both subunits on the structure of DhPceA and DhPceB, respectively. Weblogo[63] was used to highlight selected regions of the RdhA and RdhB sequence alignments.

## Reporting summary
Further information on research design is available in the Nature Portfolio Reporting Summary linked to this article.

## Data availability
The atomic model and cryo-EM map obtained for the PceA$_2$B$_2$ complex were deposited to wwpdb as PDB under accession code 8Q4H and EMDB under accession code EMD-18148, respectively. EM raw data were deposited to EMPIAR under accession code EMPIAR-11719. The LC-MS/MS data were deposited to Zenodo under accession code 10042884. Requests for material should be addressed to the corresponding authors. Source data are provided with this paper.

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

## Acknowledgements

We thank the Dubochet Center for Imaging (DCI) in Lausanne for cryo-EM data collection. DCI is an initiative of the EPFL, University of Lausanne and University of Geneva. We also thank Kelvin Lau from PTPSP laboratory (EPFL) and the PCF laboratory (EPFL). This work was supported by the Swiss National Science Foundation (grant N° CRSII5_177195 and grant N° 310030_188548 to HS; grant N° 31003 A 173059 to CH), the Sino-Swiss Science and Technology Cooperation SSSTC (N° IZLCZ0_206089 to HS), the Fundação para a Ciência e Tecnologia for support through the MOSTMICRO-ITQB unit (UIDB/04612/2020 and UIDP/04612/2020 to AGD and IACP) and the Associated Laboratory LS4FUTURE (LA/P/0087/2020 to AGD and IACP).

## Author contributions

L.C. performed all the experimental work and participated in data analysis. D.N. and B.E. performed cryo-EM experiments and its analysis. I.A.C.P., A.G.D., C.H. and J.M. supervised the experiments. J.M., L.C. and A.G.D. conceived the experiments. L.C., A.G.D. and J.M. wrote the manuscript with contributions from I.A.C.P., C.H. and H.S. All authors read and approved the manuscript.

## Competing interests

The authors declare no competing interests.
