## [Peer Review File · Nature Communications]

Structure of a membrane-bound menaquinol:organohalide oxidoreductaseREVIEWER COMMENTS

Reviewer #1 (Remarks to the Author):

The manuscript by Cimmino et al describes the cryo-EM structure of a membrane-bound PCE-dehalogenase that is involved in respiration in a *Desulfitobacterium* strain. The presented results are novel, interesting and important as they resolve a long awaited problem in the field, how the respiratory RdhA proteins are bound to RdhB proteins in the membrane. The obtained structure has a very good resolution and surprisingly also contained bound menaquinone (MK) and the product DCE.

The structure also resolves the position of 2 [4Fe-4S] clusters in the RdhA protein and the corrinoid in the active center. It detects two possible substrate channels. All this is exciting to read.

In addition to the good results, a lot of important and useful methods are described, for example the purification procedures. The bands in the CN-gel can very nicely be correlated to the RdhA2B2 tetramer and RdhA2 dimer. This is all very clear. The in-gel detection of RdhA activity is newly described in this manuscript.

The manuscript is written very precisely, well-structured, concise and clear. It was a pleasure to read.

Major points

1. the name MOOR is confusing. It claims, the enzyme is oxidizing MK. However, the activity tests (e.g in the gels) were all done with MV. Why not using quinone(analoga). I guess the authors tried them, but why not showing the results? Only then it is clear, the purified enzyme is a MOOR. The results show a methyl-viologen-dependent PCE oxidoreductase. The presence of MK shows binding but not oxidation of MK. In this way, several wordings are critical: In 17 "functional" MOOR. (no, this function has not been shown); In 250; In 154/255 "demonstrating the role of the quinone pool in electron transfer to the catalytic subunit" - the present work shows very nicely the presence of MK, not its role; also line 324/325. and In 334 "PUTATIVE MQH oxidation"

2. RdhC is clearly a part of the *rdh* operon in *Desulfitobacterium* and is also expressed. What is its role? I agree that the current manuscript focusses on the RdhAB structure and for that RdhC is not important. However, when the manuscript discusses the role of RdhAB in the respiratory chain, RdhC must be discussed, even if the authors have no explanation. Could it be that both, MK and RdhC, are necessary for activity?

3. Substrates: in this study no substrate tests were done (different electron acceptors). I do not suggest them but it was surprising to then see a large discussion on the substrate adaptation of the enzyme (266-292) and in the conclusion (336). I think these are speculations to which this work does not really contribute. Why, for example, would the enzyme not reduce DCE to VC or ethene? Why are di-eliminations discussed although no dielimination is happening in the experimental tests?

Minor points

In 23 - as the authors point out, the RdhAB is electroneutral and does not generate a pmf. Therefore, it is not really shown how energy is conserved in OHR respiration. The energy conservation (charge separation) occurs then in the hydrogenase that is not investigated here.

In 48 - proteobacteria should be included

In 74 - hydrogen is not per se a source of energy. It is the electron donor

In 124 - I guess, this is without the TAT leader, correct? Please indicate

In 164 - there is still the possibility that binding at one site changes the overall structure impacting binding at the second binding site (classical enzymatic cooperativity). Then the two sites would not be independent

In 191 - ref 37 showed that for Dehalobacter, not for Desulfobacterium (investigated here)

In 214 - 0.95 Angstrom is surprisingly narrow to me. Would PCE with the 4 Cl atoms fit through this channel? Or would protein movement be necessary?

In 230 - does this indicate the MK might have more than the 2 isoprene subunits shown in the figure?

May be indicate below the figure.

In 258/259 - under this argumentation, when no activity with only RdhA can be found because MK needs RdhA + RdhB for binding, then the authors should find activity with MK and PCE using the RdhAB that they isolated here (or in the CN gel band).

In 298/299 - for thermodynamic reasons also the reductive dehalogenation of DCE to VC or ethene should be possible. why does this not happen?

In 300 - 50% of the RdhAs in fig 1 come from bacteria that have no cytochrome b. What is "Normally"?

In 300 I suggest "organohalide reduction" instead of "organohalide respiration" here

In 302 - With "enzymatic respiratory complex" is meant the hydrogenase?

In 311/312 - to my impression, this is not true (same for In 338-340). H₂ oxidation acidifies the medium but does not contribute to pmf generation. RdhAB is still electroneutral. If protons are taken up, the cytoplasm would acidify.

Methods:

In 367 - give SI unit (at least additionally in brackets)

In 377 10:1 is this meant as protein:DDM or membrane mass:DDM?

In 386 - "2 h 30" - 2 h 30 min?

In 392-406: the procedure is new and interesting and should be described more precisely:

395 - how large is the serum flask? is the gel rolled and pressed through the narrow neck? How does this work without breaking the gel?

396 how much of the Tris buffer?

398 - how much zinc powder?

why is the gas exchange to N₂ necessary? to avoid MV reduction by hydrogenase? but this should not be at the same spot in the gel; why bringing the closed serum bottles back into the tent if they are closed?

In 437 superscript 2

Fig. 3 violet/purple instead of "lila"?

Fig. 5A here the MK interaction with RdhB is not getting clear. 5C: indicate the MK was not well resolved and might have more than two isoprene-units.

Extended figure 2b: which PTMs are you referring to here exactly?

Reviewer #2 (Remarks to the Author):

Organohalide respiration is not only of great interest for basic science, but also for application. Halogenated compounds, which can sometimes have great toxicity, accumulate in the environment mainly through human activity and can be effectively removed from there by organohalid respiration. Thus, this physiological process that takes place in bacteria plays an important role in bioremediation. Furthermore, anaerobic bacteria also use this process to build a protonmotive force (pmf) that enables them to synthesize ATP. The key enzyme in this process is a membrane-bound reductive dehalogenase that catalyzes halogen elimination or substitution. Now, Cimmino et al. report the cryo-EM structures of this enzyme from *Desulfitobacterium hafniense* at about 2.9 Å resolution.

In the manuscript, the architecture of the two proteins forming the complex, PceA and PceB, is shown as well as the arrangement of the cofactors that catalyze the reaction. The complex is arranged as a heterodimer (PceA₂B₂). The peripheral subunit PceA contains channels that enable separate diffusion of the substrate and the product to the active site. The active site is made up by a cobalt corrinoid cofactor that receives the electron for substrate reduction from two iron-sulfur cluster. Most surprisingly, a bound menaquinone molecule is found in defined position at the interface of PceA and PceB in both dimers. Altogether the data provided in the manuscript unequivocally show that the enzyme functions as a dimer with two independent pathways and that reduced menaquinone is the electron donor for reductive dehalogenation. Amino acid residues involved in de-protonation of the substrate menaquinol were identified on the P-side of the membrane. Accordingly, the contribution of reductive dehalogenation to the generation of a protonmotive force is now evident. Together with a hydrogenase (or another primary dehydrogenase in other organisms) menaquinone is reduced from the N-side of the membrane while menaquinol is oxidized by the reductive dehalogenase on the P-side. Thus, the enzyme is involved in the formation of a classic 'redox loop' that charges the membrane.

Thus, the manuscript deals with a general and highly important topic. It contains a lot of new and important information. The structures presented will for sure be an excellent guideline for many researchers in the field. The data analysis and their interpretation were performed with great care. The manuscript is written in a very clear and careful manner. There are only a few minor points that should be taken into consideration:

- I. 109: The abbreviation TCE has already been introduced in I. 49.
- I. 112: The detergent used should already be specified here and not generally referred to as a detergent.
- I. 130: The elution profile of the size-exclusion chromatography is an important piece of information that should be shown, e.g. in Fig. 2 or Extended Data Fig. 1.
- I. 231 - 233: The distances from the amino acid residues stabilizing quinone binding to the quinone headgroup should be provided in the text and in Fig. 5.
- I. 235: The distances from glutamate residues 61 and 63 to the quinone headgroup should be provided in the text and in Fig. 5
- I. 293 ff.: The discussion about the thermodynamics of the reaction is a bit misleading. The quotation [48] describes biological reaction rates but not equilibria, like thermodynamics. It would be more understandable, if [48] is simply quoted after the following sentence [.. and drive electron transfer.].
- I. 311: replace 'stoichiometry' by '... contribute to the establishment of a pmf due to the different

orientation of the quinone reduction and quinol oxidation sides relative to the membrane’.

I. 421: Describe the composition of the buffer used for size-exclusion chromatography.

Extended Data Fig. 4: The meaning of the asterisk in Fig 4a should be explained in the legend.

RESPONSES TO REVIEWER COMMENTS

Reviewer #1 (Remarks to the Author):

The manuscript by Cimmino et al describes the cryo-EM structure of a membrane-bound PCE-dehalogenase that is involved in respiration in a *Desulfitobacterium* strain. The presented results are novel, interesting and important as they resolve a long-awaited problem in the field, how the respiratory RdhA proteins are bound to RdhB proteins in the membrane. The obtained structure has a very good resolution and surprisingly also contained bound menaquinone (MK) and the product DCE.

The structure also resolves the position of 2 [4Fe-4S] clusters in the RdhA protein and the corrinoid in the active centre. It detects two possible substrate channels. All this is exciting to read.

In addition to the good results, a lot of important and useful methods are described, for example the purification procedures. The bands in the CN-gel can very nicely be correlated to the RdhA₂B₂ tetramer and RdhA₂ dimer. This is all very clear. The in-gel detection of RdhA activity is newly described in this manuscript.

The manuscript is written very precisely, well-structured, concise and clear. It was a pleasure to read.

Major points

1. the name MOOR is confusing. It claims, the enzyme is oxidizing MK. However, the activity tests (e.g in the gels) were all done with MV. Why not using quinone (analoga). I guess the authors tried them, but why not showing the results? Only then it is clear, the purified enzyme is a MOOR. The results show a methyl-viologen-dependent PCE oxidoreductase. The presence of MK shows binding but not oxidation of MK. In this way, several wordings are critical: In 17 "functional" MOOR. (no, this function has not been shown); In 250; In 154/255 "demonstrating the role of the quinone pool in electron transfer to the catalytic subunit" - the present work shows very nicely the presence of MK, not its role; also line 324/325. and In 334 "PUTATIVE MQH oxidation"

Response:

We agree with the reviewer that we do not have unambiguous evidence of a functional menaquinol-oxidising activity of the PceA₂B₂ complex. In the manuscript, we have now nuanced our statements to reflect this view, following the reviewer's comments.

We have tried electron-transfer experiments monitored by UV-Visible spectroscopy using menadiol as the electron donor and PCE as the terminal electron acceptor without success. This activity was tested with the crude cell and membrane extracts. The major challenge is likely to be the apparently unstable nature of the PceA₂B₂ complex in solution, as evidenced by the very low portion of particles observed in the cryo-EM analysis displaying the tetrameric complex.

Nevertheless, the presence of MK at its binding site at the interface of PceA and PceB, in close vicinity to the distal FeS cluster of PceA, is a very strong evidence for its involvement in electron transfer to the PceA active site and thus a role as electron donor, as also stated by Reviewer 2. We believe that the MOOR acronym (menaquinol:organohalide oxidoreductase) represents the best

term to describe our results and that this role is very likely to be widespread across MK-dependent OHRB. However, according to the reviewer comment we made several changes in the manuscript (line numbers indicate the position in the revised manuscript):

- l. 18: the term 'functional' was removed from the sentence.
- l.260: the term 'MOOR' was removed from the sentence.
- l. 257-8: this sentence was kept unchanged as the statement regarding the role of quinone originates from previous studies, as referenced in the text.
- l. 333-4: the text was changed to '(...) the minimal respiratory RDase module that is most likely capable of delivering electrons from MKH₂ (...)'.
(...)
- l.344-5: the text was changed to '(...) and together form a quinone binding site. Menaquinol is likely oxidised on the P-side of the membrane (...)'

2. RdhC is clearly a part of the *rdh* operon in *Desulfitobacterium* and is also expressed. What is its role? I agree that the current manuscript focusses on the RdhAB structure and for that RdhC is not important. However, when the manuscript discusses the role of RdhAB in the respiratory chain, RdhC must be discussed, even if the authors have no explanation. Could it be that both, MK and RdhC, are necessary for activity?

Response:

A few sentences about PceC (or generally RdhC) were added to the manuscript (from l. 60 of the revised version). All evidence so far suggests that it is not required for electron transfer, namely: (i) while *pceC* is part of the *pce* operon in *D. hafniense* strain TCE1 and in other OHRB, it is not universally present in OHRB (Book chapter by Kruse et al, 2016; Buttet et al, 2018), and also not in all *rdh* gene clusters from MK-dependent OHRB; (ii) the first proposition for the function of PceC (Maillard *et al.*, 2005), in analogy to CprC of *Desulfitobacterium dehalogenans* (Smidt *et al.*, 2000), was a role as regulator; (iii) in a previous work (Cimmino *et al.*, 2022), quantitative proteomics were applied to evaluate the stoichiometry of PceA, PceB, PceC and PceT in both *D. hafniense* strain TCE1 and *D. restrictus*. There, while PceA and PceB were identified with a ~2:1 stoichiometry, PceC was detected with a 50-fold lower abundance than PceA.

These statements and results indicate that the role of RdhC is not essential for OHR, and so it is not strange that it is not part of the RDase complex. So far there is no concrete evidence for its possible function.

Added text in the revised manuscript:

- l. 62-65: the following text was added to the manuscript: '(...), while PceC was detected at a 50× lower abundance than PceA¹⁸. These results, and the absence of *pceC* in many OHRB, challenged the working model for electron transfer in *D. restrictus* that we have proposed in 2016, where PceC could be the missing redox protein between the quinol pool to PceA¹⁹.'

3. Substrates: in this study no substrate tests were done (different electron acceptors). I do not suggest them but it was surprising to then see a large discussion on the substrate adaptation of the enzyme (266-292) and in the conclusion (336). I think these are speculations to which this work does not really contribute. Why, for example, would the enzyme not reduce DCE to VC or ethene? Why are di-eliminations discussed although no di-elimination is happening in the experimental tests?

Response:

The overall idea of the paragraph discussing the substrate specificity of DhPceA was to try to find structural basis for this specificity, namely involving the predicted substrate channels, and to

compare these with those of *S. multivorans* PceA (Bommer *et al*, 2014), an enzyme that is more permissive for bulkier substrates than DhPceA. On the same line, literature data already suggested that 3-Cl-4-OH-phenylacetate, a well-studied aromatic compound in OHR is not dechlorinated by DhPceA (Gerritse *et al*, 1999).

The lack of dechlorination of DCE and VC by DhPceA has been shown, but it cannot be explained by the size of the channel, nor by the redox potential. *D. restrictus* PceA, which shares 99% sequence identity with DhPceA was clearly shown to be incapable of dechlorinating DCE isomers (Maillard *et al*, 2003). No obvious explanation can be driven from the structure of the enzyme. It appears to be a common theme in several PCE-dechlorinating enzymes that the reaction stops at DCE. Maybe the orientation of DCE in the active site does not allow the electron transfer from reduced cobalamin to the substrate.

We do not really discuss dihaloelimination in our manuscript. The only indirect mention to it is the DcaA enzyme from *D. dichloroeliminans* which shows 88% sequence identity to DhPceA, and to a variant thereof that confers a slightly different substrate specificity, as reported by (Kunze *et al*, 2017).

Minor points

In 23 - as the authors point out, the RdhAB is electroneutral and does not generate a pmf. Therefore, it is not really shown how energy is conserved in OHR respiration. The energy conservation (charge separation) occurs then in the hydrogenase that is not investigated here.

Response:

We believe that the reviewer refers to l. 307 and 338 in the manuscript where we mention that the RdhAB complex is electroneutral.

Although the PceAB complex is classified as electroneutral, as it releases electrons and protons on the same side of the membrane (P-side), it contributes to energy conservation by being involved in a redox loop with the hydrogenase (or an alternative dehydrogenase), as shown in Figure 1. Thus, the PceAB complex contributes to and is essential for the formation of a *pmf*. This was now more clearly explained in the text (l. 317-22).

In 48 - proteobacteria should be included

Response:

We have now included the Proteobacteria with key representative *Sulfurospirillum* in the manuscript. They belong to the group of quinone-dependent OHRB.

Added text in the revised manuscript:

- l. 48: '(...), and to the Proteobacteria such as *Sulfurospirillum* (...)'

In 74 - hydrogen is not per se a source of energy. It is the electron donor

Response:

In H₂-oxidising chemolithotrophic anaerobic bacteria, hydrogen is both the electron donor and the source of energy. Furthermore, it has recently been shown that aerobic bacteria can also extract energy from atmospheric hydrogen (Grinter *et al*, 2023, Nature 615:541).

In 124 - I guess, this is without the TAT leader, correct? Please indicate

Response:

Correct. This is now mentioned (l. 129 of the revised manuscript). However, after recalculation of the theoretical molecular weight, the value has been corrected to 142.7 kDa (PceA₂B₂ including 4 [4Fe/4S] clusters and 2 cobalamins without any upper ligand).

In 164 - there is still the possibility that binding at one site changes the overall structure impacting binding at the second binding site (classical enzymatic cooperativity). Then the two sites would not be independent

Response:

This is true. Our statement comes from the description of *S. multivorans* PceA dimer (Bommer *et al*, 2014). While we believe that the distance between both active sites is too large for cooperativity, the manuscript was modified to account for the possibility of cooperative binding sites.

Modified text in the revised manuscript:

- l. 168-71: '(...), which suggests that the complex has two independent catalytic sites, one per heterodimer, as it was already proposed for *S. multivorans* PceA dimer. However, possible cooperativity between both binding sites cannot be excluded.'

In 191 - ref 37 showed that for Dehalobacter, not for Desulfitobacterium (investigated here)

Response:

The reviewer is right. However, the very high sequence identity (99%) between PceA from *D. restrictus* and *D. hafniense* strain TCE1 allows us to consider that the EPR data obtained from *D. restrictus* are also valid for strain TCE1. Nevertheless, this was now made more accurate in the revised manuscript. Due to changes in the manuscript, the corresponding reference is now n° 38.

Modified text in the revised manuscript:

- l. 197: 'At the PceA active site of *D. restrictus* (and most likely also at that of *D. hafniense* strain TCE1), there is a cobalt-containing corrinoid held in a base-off conformation, as identified earlier by EPR spectroscopy³⁸ (...).'

In 214 - 0.95 Angstrom is surprisingly narrow to me. Would PCE with the 4 Cl atoms fit through this channel? Or would protein movement be necessary?

Response:

The reviewer is right. The predicted channels display a radius of 0.95 angstroms at its narrowest section, which is quite small for the diffusion of PCE, TCE or DCE molecules. However, these channels were predicted in the structure, without any type of energy minimization or dynamic simulation, and without considering the diffusion of the molecule through the channels or its interaction with the amino acids or their side chain. The text was modified accordingly.

Modified text in the revised manuscript:

- I.221: '(...) 0.95 Å radius. While this section appears too narrow for the diffusion of chlorinated ethenes, it must be noted that this analysis is static and does not consider the dynamic nature and flexibility of the channel upon substrate transfer to and out of the active site.'

In 230 - does this indicate the MK might have more than the 2 isoprene subunits shown in the figure? May be indicate below the figure.

Response:

Correct. MK in *D. hafniense* strain TCE1 has been identified in our laboratory in the form of MK-7 (most abundant) and MK-8 (unpublished data). It suggests that MK has 7 isoprene subunits, which is likely running alongside the hydrophobic residues in the TMH2 and -3 of PceB. The text was modified accordingly in the legend of Fig. 5c.

In 258/259 - unter this argumentation, when no activity with only RdhA can be found because MK needs RdhA + RdhB for binding, then the authors should find activity with MK and PCE using the RdhAB that they isolated herein (or in the CN gel band).

Response:

Please see above the answer to the 1st major point raised by the reviewer.

In 298/299 - for thermodynamic reasons also the reductive dehalogenation of DCE to VC or ethene should be possible. why does this not happen?

Response:

The reviewer is right. There is no thermodynamic explanation to the lack of dechlorination of DCE and VC by PceA. One possible reason could be the configuration of the active site, and its secondary coordination sphere that may not allow a correct positioning of the chlorine atom for DCE or VC dechlorination.

In 300 - 50% of the RdhAs in fig 1 come from bacteria that have no cytochrome b. What is "Normally"?

Response:

'Normally' was removed and the sentence was slightly modified.

Modified text in the revised manuscript:

- I. 309: 'MK-dependent OHRB couple H₂ oxidation (...)'

In 300 I suggest "organohalide reduction" instead of "organohalide respiration" here

Response:

Correct. This was modified according to the reviewer's comment (I. 309).

In 302 - With "enzymatic respiratory complex" is meant the hydrogenase?

Response:

Correct. The sentence and the description of the electron transfer process was modified to make it clearer (l. 311-22 of the revised manuscript).

In 311/312 - to my impression, this is not true (same for ln 338-340). H₂ oxidation acidifies the medium but does not contribute to pmf generation. RdhAB is still electroneutral. If protons are taken up, the cytoplasm would acidify.

Response:

In MK-dependent OHRB, the membrane bound hydrogenases are electrogenic and contribute to the formation of a *pmf* in a redox loop together with the PceAB complex, as discussed above. The key subunit in the hydrogenase is a cytochrome *b* that take up electrons from hydrogen oxidation (P-side) and take up protons from the cytoplasm (N-side). Then the PceAB complex releases the protons from menaquinol oxidation to the periplasmic side of the membrane due to the position of its quinone binding site. The protons from H₂ oxidation indeed do not contribute to the *pmf*. The *pmf* is used by the ATP synthases for ATP generation.

Methods:

In 367 - give SI unit (at least additionally in brackets)

Response:

This was modified accordingly. 1000 psi = 6.9 MPa.

In 377 10:1 is this meant as protein:DDM or membrane mass:DDM?

Response:

We thank the reviewer to point out this ambiguity. The 10:1 ratio corresponds to the DDM:protein mass ratio. The text in the manuscript has been revised.

Modified text in the revised manuscript:

- l. 388: '(...) at a DDM:protein (w/w) ratio of 10:1 (...)

In 386 - "2 h 30" - 2 h 30 min?

Response:

This has been modified accordingly.

In 392-406: the procedure is new and interesting and should be described more precisely:

Response:

The text in the revised manuscript (l. 404-18) has been modified to give additional technical details. Below, extra practical details are also given to answer the specific questions of the reviewer.

395 - how large is the serum flask? is the gel rolled and pressed through the narrow neck? How does this work without breaking the gel?

Response:

It was a 500-mL serum flask. The vertical gel lanes (up to three lanes) were cut from the gel and passed through the neck of the flask in a vertical position. A bit of water was used to let the gel slide down in the flask. The gel is relatively fragile (most particularly the top that has the lowest acrylamide concentration), but by pulling it from the top with a plastic tweezer and detaching from the glass plate by flowing some water at the top end of the lane, it works.

396 how much of the Tris buffer?

Response:

~10 mL of buffer is added to the flask to equilibrate the gel lane and avoid desiccation.

398 - how much zinc powder?

Response:

MV was reduced with 1 g of zinc powder.

Why is the gas exchange to N₂ necessary? to avoid MV reduction by hydrogenase? but this should not be at the same spot in the gel

Response:

Yes, in order to avoid any undesired reduction of MV by hydrogenase activity, H₂ from the glovebox atmosphere was removed from the flask. In the setting up phase of the protocol, we indeed noticed some hydrogenase activity (red band appearing on a colourless gel), that was not overlapping with the PCE Rdase activity bands. This means that a slightly modified version of the protocol can be used for investigating hydrogenase complexes.

Why bringing the closed serum bottles back into the tent if they are closed?

Response:

In order to prevent any oxygen contamination in the flask by multiple perforations of the septum, we continued the assay by placing the flask back into the glovebox and added the remaining elements with the use of syringes, thus also avoiding any H₂ contamination in the flask from the glovebox atmosphere.

In 437 superscript 2

Response:

This was modified accordingly.

Fig. 3 violet/purple instead of "lila"?

Response:

This was changed to 'light purple'.

Fig. 5A here the MK interaction with RdhB is not getting clear. 5C: indicate the MK was not well resolved and might have more than two isoprene-units.

Response:

Fig. 5a was designed to show the helix bundle arrangement around the MK. Fig. 5c was designed to show the contact between MK and key residues in PceB.

Fig. 5c: indeed, MK was only resolved with two isoprene units in the cryo-EM density map. It is indicated now in the legend of Fig. 5c. It is most likely that MK-7 or MK-8 are delivering electrons to

the complex. At least, these two forms have been identified in the past as major MK molecules from cell extracts of *D. hafniense* strain TCE1 (unpublished data).

Extended figure 2b: which PTMs are you referring to here exactly?

Response:

PTMs in Extended Data Fig. 2b are indicated by default in the raw data of proteomics. It corresponds to peptides for which a certain number of spectra were obtained with PTMs (oxidation for Met, and phosphorylation for Ser, Thr and Tyr).

Modified text in the revised Extended Data Material:

'Green residues indicate amino acids that were found post-translationally modified for a certain number of the detected peptides (oxidation for Met; phosphorylation for Ser, Thr and Tyr).'

Reviewer #2 (Remarks to the Author):

Organohalide respiration is not only of great interest for basic science, but also for application. Halogenated compounds, which can sometimes have great toxicity, accumulate in the environment mainly through human activity and can be effectively removed from there by organohalide respiration. Thus, this physiological process that takes place in bacteria plays an important role in bioremediation. Furthermore, anaerobic bacteria also use this process to build a protonmotive force (pmf) that enables them to synthesize ATP. The key enzyme in this process is a membrane-bound reductive dehalogenase that catalyzes halogen elimination or substitution. Now, Cimmino et al. report the cryo-EM structures of this enzyme from *Desulfitobacterium hafniense* at about 2.9 Å resolution.

In the manuscript, the architecture of the two proteins forming the complex, PceA and PceB, is shown as well as the arrangement of the cofactors that catalyze the reaction. The complex is arranged as a heterodimer (PceA₂B₂). The peripheral subunit PceA contains channels that enable separate diffusion of the substrate and the product to the active site. The active site is made up by a cobalt corrinoid cofactor that receives the electron for substrate reduction from two iron-sulfur cluster. Most surprisingly, a bound menaquinone molecule is found in defined position at the interface of PceA and PceB in both dimers. Altogether the data provided in the manuscript unequivocally show that the enzyme functions as a dimer with two independent pathways and that reduced menaquinone is the electron donor for reductive dehalogenation. Amino acid residues involved in de-protonation of the substrate menaquinol were identified on the P-side of the membrane. Accordingly, the contribution of reductive dehalogenation to the generation of a protonmotive force is now evident. Together with a hydrogenase (or another primary dehydrogenase in other organisms) menaquinone is reduced from the N-side of the membrane while menaquinol is oxidized by the reductive dehalogenase on the P-side. Thus, the enzyme is involved in the formation of a classic 'redox loop' that charges the membrane.

Thus, the manuscript deals with a general and highly important topic. It contains a lot of new and important information. The structures presented will for sure be an excellent guideline for many researchers in the field. The data analysis and their interpretation were performed with great care. The manuscript is written in a very clear and careful manner. There are only a few minor points that should be taken into consideration:

I. 109: The abbreviation TCE has already been introduced in I. 49.

Response:

This was modified accordingly (I. 114).

I. 112: The detergent used should already be specified here and not generally referred to as a detergent.

Response:

This was modified accordingly (I. 117-8).

I. 130: The elution profile of the size-exclusion chromatography is an important piece of information that should be shown, e.g. in Fig. 2 or Extended Data Fig. 1.

Response:

The elution profile of the size-exclusion chromatography was added to Extended Data Fig. 1 (new panel c).

I. 231 - 233: The distances from the amino acid residues stabilizing quinone binding to the quinone headgroup should be provided in the text and in Fig. 5.

Response:

The distances are provided now in the manuscript and Fig. 5b.

I. 235: The distances from glutamate residues 61 and 63 to the quinone headgroup should be provided in the text and in Fig. 5

Response:

The distances between E61 and E63 to the quinone headgroup are between 8 and 13 Å. However, these residues are unlikely to interact directly with the quinone and are more likely to participate in a proton channel together with other conserved residues (H495, D515 and Y520). E61 and E63 may represent the exit gate for protons to the bulk liquid on the P side of the membrane.

I. 293 ff.: The discussion about the thermodynamics of the reaction is a bit misleading. The quotation [48] describes biological reaction rates but not equilibria, like thermodynamics. It would be more understandable, if [48] is simply quoted after the following sentence [.. and drive electron transfer.].

Response:

Ref. 48 (now Ref. 49 in the revised manuscript) has been placed according to the reviewer's comment (l. 308 of the revised manuscript).

I. 311: replace 'stoichiometry' by '... contribute to the establishment of a pmf due to the different orientation of the quinone reduction and quinol oxidation sides relative to the membrane'.

Response:

This sentence was replaced with a more detailed description of the process, also in answer to Reviewer 1, and the position of the two quinone binding sites in the cytoplasmic membrane is mentioned (l. 311-22 of the revised manuscript).

I. 421: Describe the composition of the buffer used for size-exclusion chromatography.

Response:

The composition of Buffer 3 was already given in the previous paragraph of the manuscript (l. 423 of the revised manuscript). Buffer is composed of 25 mM HEPES-NaOH pH: 7.5, 150 mM NaCl, 0.005% Lauryl Maltose Neopentyl Glycol (LMNG).

Extended Data Fig. 4: The meaning of the asterisk in Fig 4a should be explained in the legend.

Response:

We thank the reviewer to make us aware of this omission. We have added the following sentence to the legend in Extended Data Fig. 4a: 'The hash sign, the cubes and the asterisk indicate the position of the cofactors in the PceAB dimer: the cobalamin, both [4Fe-4S] clusters and the menaquinone, respectively.'

REVIEWERS' COMMENTS

Reviewer #1 (Remarks to the Author):

The review has been done with care and I don't have further objections to publication.